# The transcription factor *TaWRKY58* coordinates growth and drought sensitivity in wheat by repressing *TaLRR* and *TaBCS1*

Yazhou Zhang[1,2]*, Xinyao Cheng[2]*, Xinyu Yu[2], Anyu Gu[2], Xufei Zhao[2], Mei Deng[2], Guoyue Cheng[1,2], Qiang Xu[1,2], Qiantao Jiang[1,2], Yuming Wei[1,2]*

1 State Key Laboratory of Crop Gene Exploration and Utilization in Southwest China, Sichuan Agricultural University, Chengdu, Sichuan, China, 2 Triticeae Research Institute, Sichuan Agricultural University, Chengdu, Sichuan, China

* yazhou14716@sicau.edu.cn (YZ); 15239799635@163.com, ymwei@sicau.edu.cn (YW)

## Abstract

Balancing growth and stress adaptation is essential for optimizing crop productivity, yet the transcriptional mechanisms underlying this trade-off in wheat remain poorly understood. Here, we identify the WRKY transcription factor *TaWRKY58* as a key repressor coordinating plant architecture and drought response. Loss-of-function mutants of *TaWRKY58* exhibit increased plant height and early flowering, accompanied by elevated gibberellin levels, while overexpression restores wild-type phenotypes. Under drought stress, *TaWRKY58* represses soluble sugar accumulation, and its mutants show enhanced drought tolerance. Using DAP-seq, we identified genome-wide binding sites of TaWRKY58 and uncovered a W-box-like motif enriched in its target promoters. Electrophoretic mobility shift and dual-luciferase assays confirmed that TaWRKY58 directly binds to and represses two key targets: *TaLRR*, encoding a leucine-rich repeat protein, and *TaBCS1*, encoding a mitochondrial AAA$^+$ ATPase. Mutants of *TaLRR* and *TaBCS1* display dwarfism and drought hypersensitivity, respectively, mirroring aspects of the *TaWRKY58* overexpression phenotype. Our data support a model in which *TaWRKY58* functions as a transcriptional repressor in a coherent regulatory module that fine-tunes growth and stress adaptation by modulating signaling and energy metabolism. This mechanism offers a potential strategy for breeding wheat with optimized yield stability under fluctuating environments.

## Author summary

Wheat need balance growth and stress survival to maintain yield. We show that the TaWRKY58 protein acts as a central regulator of this balance. It functions as a repressor, reducing plant height by limiting growth hormones and curbing drought responses. We identified two key genes—*TaLRR* (signaling) and *TaBCS1* (energy)—that TaWRKY58 directly turns off. Mutating these genes

**Data availability statement:** All relevant data are included within the article and its supplementary materials.

**Funding:** This work was supported by the National Key Research and Development Program-Creation and Application of New Disease-Resistant Wheat Germplasm in the Southwest Wheat-growing Area (2024YFD1201202 to Y.Z.Z.), and The Sichuan Provincial Agricultural Department Innovative Research Team (SCCXTD-2024-11 to Y.M.W.). The funders had no role in study design, data collection and analysis, decision to publish, or preparation of the manuscript.

**Competing interests:** The authors have declared that no competing interests exist.

caused dwarfism or drought sensitivity, confirming their role in this trade-off. Our findings reveal a simple genetic module that coordinately controls plant architecture and stress tolerance, offering new targets for breeding more resilient wheat.

## Introduction

Wheat is a globally vital food crop, providing approximately 20% of the world's dietary calories and serving as the primary calorie source for humans. Its importance stems from strong environmental adaptability, high yield, high energy density, ease of storage, and diverse processing applications. To withstand diverse biotic and abiotic stresses during growth, wheat has evolved a sophisticated adaptive defense system involving physical barriers, signaling networks, metabolic reorganization, and gene reprogramming. Transcription factors (TFs) act as key "regulators" within this system, potentially enabling the identification of an optimal equilibrium between growth and stress resistance. Achieving this balance is critical for achieving the breeding objectives of high yield under optimal conditions and stable yield under stress [1,2]. Consequently, elucidating the regulatory mechanisms of TFs in wheat offers promising targets for optimizing multiple superior agronomic traits through genetic engineering, thereby enhancing food security.

WRKY transcription factors TFs constitute one of the ten largest TF families in plants. They are characterized by a signature WRKY domain—containing the highly conserved WRKYGQK heptapeptide—at the N-terminus, and a zinc finger motif ($C_2$-$H_2$ or $C_2$-HC) at the C-terminus [3,4]. Based on the number of WRKY domains and the type of zinc finger motif, WRKY proteins are classified into three primary groups (I, II, and III), with further subdivisions into subgroups depending on whether one or two WRKY domains reside within a ~60-amino acid N-terminal region, along with the conserved heptapeptide sequence and a C-terminal zinc finger-like motif [3,5]. The W-box (YTGACY) in target gene promoters represents the minimal consensus sequence required for binding WRKY domains [4]. Functionally characterized WRKY TFs act predominantly as negative regulators, with fewer acting as positive regulators. They orchestrate the expression of multiple genes involved in diverse processes, including plant growth and responses to abiotic and biotic stresses [3,6,7].

Over 130 TaWRKY proteins have been identified in wheat [3]. Accumulating evidence indicates that WRKY TFs modulate plant hormone signaling pathways, acting as positive or negative regulators in wheat responses to biotic and abiotic stresses. For instance, *TaWRKY2*, *TaWRKY19*, and *TaWRKY93* enhance salinity and drought tolerance by promoting osmoprotectant accumulation (proline and soluble sugars), thereby increasing grain yield [8,9]. *TaWRKY49* negatively regulates, whereas *TaWRKY62* positively regulates, wheat seedling resistance to *Puccinia striiformis* f. sp. *tritici* under high-temperature stress via differential modulation of salicylic acid (SA)-, jasmonic acid-, ethylene-, and reactive oxygen species-mediated signaling [7]. Virus-induced gene silencing (VIGS) of *TaWRKY17* reduces salt tolerance by altering the expression of abscisic acid (ABA)/ROS-related and stress-response

genes, consequently compromising antioxidant capacity [10]. Despite these advances, our understanding of the extensive repertoire of WRKY TFs in wheat remains limited relative to their functional diversity. Therefore, further elucidation of the mechanisms underlying WRKY TF function in wheat is critically important.

In a previous study, we identified the WRKY transcription factor *TraesCS1A02G070400* (*TaWRKY58*) as being involved in ultraviolet (UV) response, SA biosynthesis, and Fusarium head blight (FHB) resistance [11]. Although this gene (also designated *TaWRKY39* or *TaWRKY53*) has been implicated in wheat heat stress response and resistance to *Puccinia striiformis* f. sp. *tritici* [12,13], its precise molecular function and regulatory mechanisms remain uncharacterized. In this study, we demonstrate that *TaWRKY58* acts as a negative regulator of gibberellin (GA) biosynthesis, modulates plant height and drought stress responses. These findings expand our understanding of WRKY TF functions and provide valuable targets for breeding wheat cultivars with enhanced agronomic traits.

## Results

### Bioinformatic analysis of *TaWRKY58*

To characterize *TaWRKY58* bioinformatically, we performed an Ensembl BLAST analysis, which revealed that *TaWRKY58* belongs to the WRKY transcription factor family and is located on chromosome 1A in wheat. Domain prediction and phylogenetic analyses indicated that TaWRKY58 contains two WRKY domains and one zinc finger motif, classifying it into subgroup Ia [3] (Fig 1A and S1 Fig).

Previous studies have suggested that *TaWRKY58* may be involved in regulating wheat responses to biotic and abiotic stresses [11–13]. Analysis of *TaWRKY58* transcriptional expression during wheat development and under various

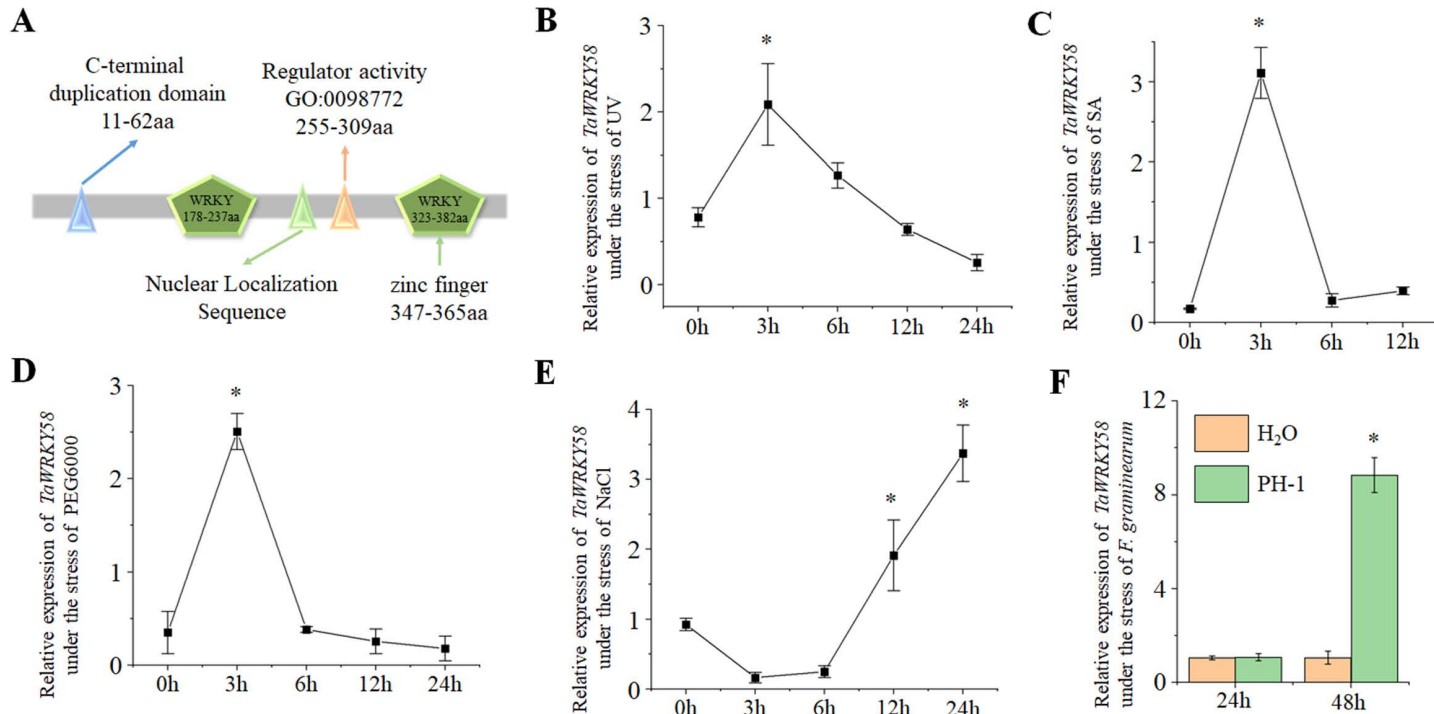

**Fig 1. Characterization of *TaWRKY58* expression in wheat. A.** Domain architecture of *TaWRKY58*. **(B–F)** Relative expression levels of *TaWRKY58* in response to: **(B)** UV radiation, (C) 10 mM SA, (D) 20% (w/v) PEG6000, (E) 200 mM NaCl, and **(F)** *F. graminearum* infection (48 hpi). Asterisks (*) above bars indicate significant differences (P ≤ 0.05) relative to controls (Student's t-test). Three biological repeats were performed for each mutant.

treatments revealed that it is expressed in multiple wheat tissues and is induced by biotic and abiotic stresses (S2 Fig). To further validate these findings, we quantified *TaWRKY58* expression via qRT-PCR in the 'Kronos' background. The results confirmed that *TaWRKY58* is significantly induced by UV radiation, SA, PEG6000, NaCl, and *Fusarium graminearum* infection (Fig 1B–1F). Notably, treatments with UV, SA, and PEG6000 elicited a transient induction of *TaWRKY58* expression (Fig 1B–1D). These results suggest that *TaWRKY58* may have a broad role in mediating wheat responses to both biotic and abiotic stresses.

### Nuclear localization of TaWRKY58

Bioinformatic analysis predicted nuclear localization for TaWRKY58 (Fig 2A). To experimentally validate this, the coding sequence of *TaWRKY58* (excluding its stop codon) was fused in-frame to the 5'-terminus of *GFP* in the pCAMBIA 1300-*GFP* vector, driven by the Ubiquitin (UBI) promoter. Confocal microscopy revealed diffuse GFP fluorescence throughout cells expressing UBI::*GFP* (control), whereas cells expressing UBI::*TaWRKY58-GFP* exhibited GFP fluorescence exclusively co-localized with the nuclear marker, confirming nuclear localization (Fig 2B). These findings provide additional evidence supporting the nuclear localization of TaWRKY58 in the nuclear.

### *TaWRKY58* Regulates plant height in wheat

Screening of the *Triticum turgidum* cv. Kronos TILLING population identified a premature stop codon (T4-2223, C1593T) in *TaWRKY58* (Fig 3A and 3B). Sanger sequencing confirmed that this mutation results in truncation of >50% of the TaWRKY58 protein (Fig 3B). To minimize background mutations, the homozygous Δ*tawrky58* mutant was backcrossed twice to the wild-type (WT) Kronos parent. We developed dCAPS markers (dCAPs-*TaWRKY58*-F/R) to genotype homozygous mutants (S3 Fig and S1 Table), selecting WT and Δ*tawrky58* lines from BC$_2$F$_6$ populations for subsequent analyses. At the Zadoks growth stage 60, Δ*tawrky58* mutants exhibited significantly increased plant height and earlier flowering compared to WT (Fig 3C and 3D). To validate this phenotype, we overexpressed *TaWRKY58* the coding sequence (driven by the UBI promoter in pCAMBIA1300) in the Δ*tawrky58* background. Ten independent transgenic lines were confirmed using UBI-specific primers, with three lines (O-*TaWRKY58*-16/20/21) showing significantly elevated *TaWRKY58* expression versus WT (Fig 3E). Notably, these overexpression lines restored WT plant height (Fig 3F and 3G).

Given GA regulates plant growth, we quantified GA levels in spikelets and rachises. The Δ*tawrky58* mutants exhibited significantly increased GA levels compared to the WT (Fig 3H). The overexpression line O-*TaWRKY58*–16 restored GA

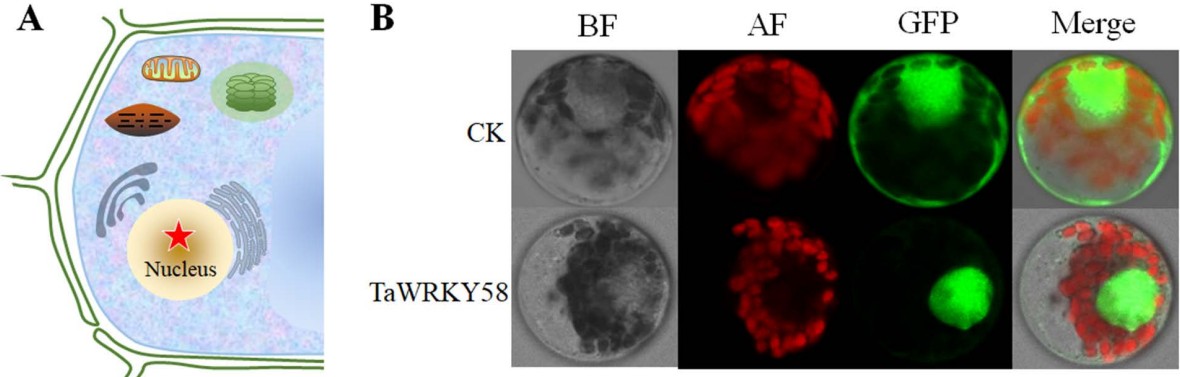

**Fig 2. Nuclear localization of TaWRKY58. A.** Schematic prediction of TaWRKY58 subcellular localization. The red five-pointed star represents the predicted location of the TaWRKY58 protein. **B.** Confocal microscopy images of wheat protoplasts expressing UBI::*TaWRKY58-GFP*. "BF" refers to brightfield microscope, "GFP" represents the fluorescence signal of GFP, and "AF" stands for the fluorescence signal of chloroplast. Scale bar: 20 μm.

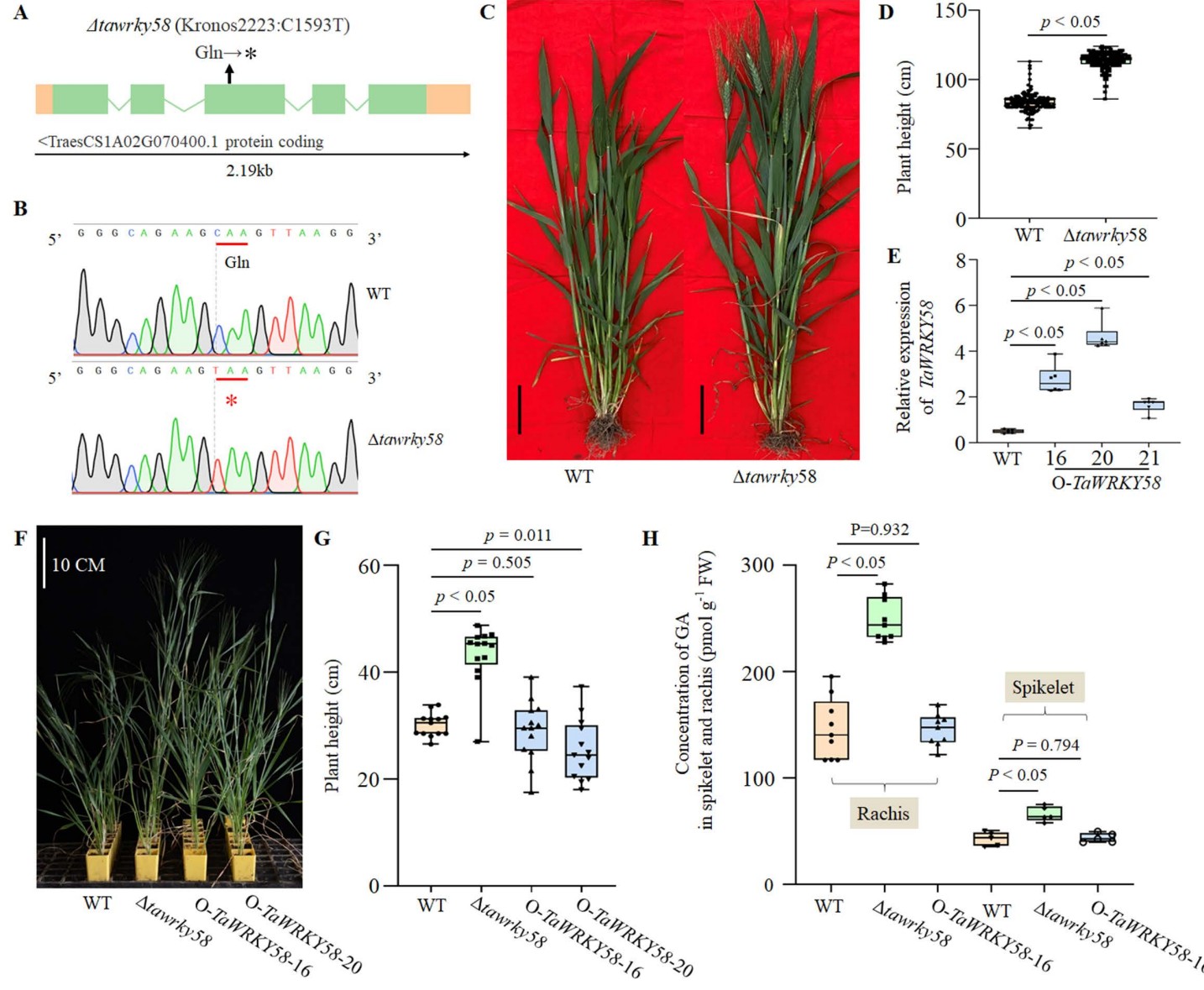

**Fig 3. Phenotypic effects of the *TaWRKY58* mutation in wheat. A.** Gene structure of wild-type *TaWRKY58* and the position of the premature stop codon (T4-2223, C1593T) in Δ*tawrky58*. **B.** Sanger sequencing chromatogram confirming the Δ*tawrky58* mutation. **C.** Plant phenotypes at Zadoks stage 60. Scale bar: 10 cm. **D.** Plant height measurements at Zadoks stage 65 for wild-type (WT) and Δ*tawrky58* homozygous lines (n ≥ 10 plants per genotype). **E.** Relative *TaWRKY58* expression in transgenic lines versus WT at Zadoks stage 13 (Z13), six biological replicates were performed for each plant. **F.** Greenhouse phenotypes of different mutations at Z60. Scale bar: 10 cm. **G.** Plant height measurements of different mutations at Z65 under greenhouse conditions. **H.** GA content in spikelets and rachises of different mutations at Zadoks stage 60. All data are presented as the mean ± SD unless otherwise stated (n ≥ 3 biological replicates).

levels to those observed in WT plants (Fig 3H). These results suggest that *TaWRKY58* negatively regulates plant height by suppressing GA accumulation.

## *TaWRKY58* Negatively regulates drought stress responses

To assess the role of *TaWRKY58* in stress responses, we first evaluated FHB susceptibility in field-grown WT and Δ*tawrky58* plants. The Δ*tawrky58* mutants showed no significant difference in FHB susceptibility compared to WT

(Fig 4A and 4B). Consistent with this, transgenic overexpression lines (O-*TaWRKY58*) exhibited FHB susceptibility similar to WT under greenhouse conditions (Fig 4C). Although Δ*tawrky58* spikelets showed a non-significant trend toward increased SA accumulation upon *F. graminearum* infection (Fig 4D), these results collectively indicate that *TaWRKY58* does not substantially contribute to FHB resistance.

We next examined the role of *TaWRKY58* in abiotic stress responses. The Δ*tawrky58* mutants did not display altered sensitivity to salt stress compared to WT and O-*TaWRKY58*–16 plants (Fig 4E and 4F). Notably, under PEG6000-induced drought stress, Δ*tawrky58* leaves accumulated significantly higher levels of soluble sugars (Fig 4G). This metabolic phenotype was further corroborated under prolonged drought treatment conditions, where Δ*tawrky58* plants also exhibited significantly reduced plant height and an earlier flowering period (Fig 4H and 4I). Additionally, more living cells were observed (S5 Fig). Concurrently, GA content was significantly enhanced in the Δ*tawrky58* mutants under these conditions (Fig 4J). Taken together, these findings demonstrate that *TaWRKY58* functions as a key negative regulator of drought stress responses in wheat, primarily by modulating hormone homeostasis.

## Genome-wide Identification of TaWRKY58 Target Genes by DAP-seq

To elucidate *TaWRKY58*-mediated regulatory mechanisms, we performed DNA affinity purification sequencing (DAP-seq) to identify genome-wide binding sites. After stringent filtering (q-value < 0.05), we identified 754,059 high-confidence TaWRKY58-binding peaks (S2 Table). The vast majority (93%) of these peaks were located in genic regions, and 7% to intergenic regions (Fig 5A). Within genic regions, TaWRKY58 exhibited the strongest binding enrichment in proximal promoters (≤1 kb upstream of transcription start sites [TSS]) (Fig 5A and 5B).

De novo motif analysis revealed Motif1 as the most significantly enriched sequence (motif score ≥5) in TaWRKY58-bound peaks, showing normal distribution characteristics (Fig 5C and 5D). We identified 4,969 putative target genes harboring Motif1 in their proximal promoters (≤1 kb upstream of TSS) (S3 Table). Gene ontology (GO) enrichment analysis indicated that these genes were predominantly associated with molecular functions, including protein serine kinase activity, protein serine/threonine kinase activity, and carbohydrate binding, etc (Fig 5E). These results establish *TaWRKY58* as a global transcriptional regulator with a broad influence on gene networks in wheat.

## *TaWRKY58* functions as a transcriptional repressor

Based on GO analysis of serine/threonine kinase activity and the selection criteria (≥2 Motif1 sites in the promoter mid-region; no homoeologs; score ≥20), we prioritized two candidate targets: *TraesCS5A02G528800* (*TaLRR*; encoding a leucine-rich repeat protein) and *TraesCS7B02G342600* (*TaBCS1*; encoding a mitochondrial AAA⁺ ATPase) (S3 Table). qRT-PCR analysis further revealed that the expression levels of *TaLRR* and *TaBCS1* were significantly upregulated in the rachis tissues of the Δ*tawrky58* mutant (Fig 6A).

To experimentally validate the binding of TaWRKY58 to the promoter regions of these genes, we designed EMSA probes based on Motif1. For the *TaLRR* promoter, a probe encompassing two variant sites (MEME-2 and MEME-3) was constructed. Similarly, for the *TaBCS1* promoter, a probe covering three variant sites (MEME-1, MEME-2, and MEME-4) was designed (Fig 6B). AlphaFold modeling predicted interaction with the canonical W-box (YTGACY), consistent with the Probe 2–1 mutation disrupting 'GT' core residues (Fig 6C). These promoter segments were subsequently used as probes in EMSA to assess TaWRKY58 binding.

EMSA results showed that the promoters of both *TaLRR* and *TaBCS1* can be bound by TaWRKY58 (Fig 6D and 6E). Moreover, a 10× competitive probe partially displaced the migration band of probe 2 and protein TaWRKY58, while a 50× competitive probe completely displaced this band (Fig 6D and 6E). Further, based on TaBCS1 probe 2, mutant probes were designed for EMSA validation. The results indicated that mutant probe 2–1 did not produce a migration band with TaWRKY58, whereas mutant probes 2–2, 2–3, and 2–4 all formed migration bands with TaWRKY58 (Fig 6E).

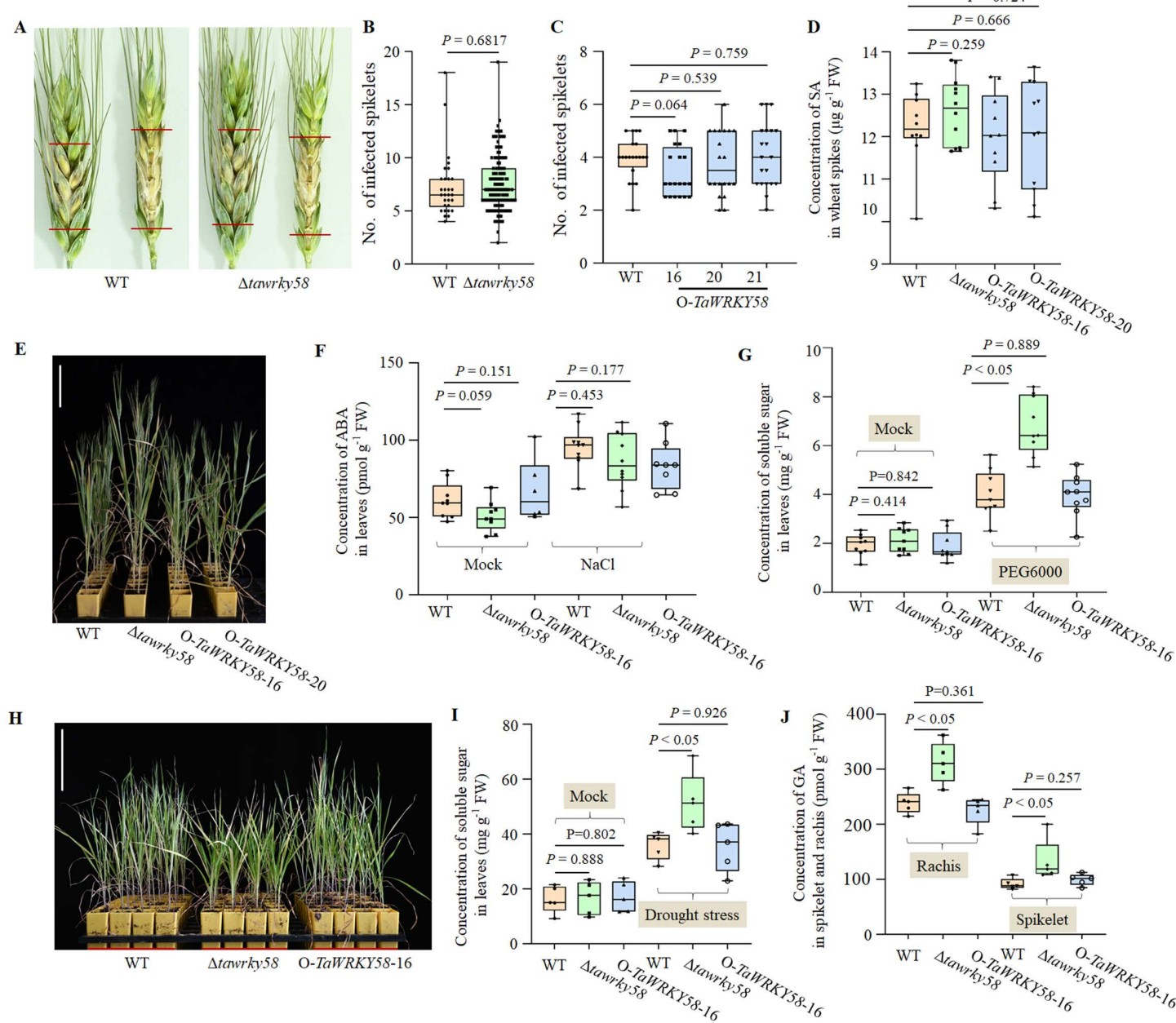

**Fig 4. Effect of *TaWRKY58* on biotic and abiotic stress in wheat. A.** Symptoms of FHB in spikes at 12 days after inoculation with *F. graminearum*. Red lines mark the extent of symptom spread. **B.** Number of infected spikelets per spike at 12 days post-inoculation with *F. graminearum*. Each mutant line was inoculated using over 30 spikes. **C.** Number of infected spikelets at 10 days after inoculation. Each transgenic line was inoculated with 20 spikes. **D.** SA accumulation in wheat. Samples were collected 48 hours post-inoculation at Zadoks growth stage 65; water-inoculated plants served as the control. FW, fresh weight. **E.** Phenotypic responses of different genotypes to 200 mM NaCl treatment for 30 days at Zadoks stage 65. Scale bar: 10 cm. **F.** ABA content in leaves of plants subjected to 200 mM NaCl for 7 days at Zadoks stage 13; water-treated plants were used as control. **G.** Soluble sugar accumulation in leaves after 7 days of 20% PEG6000 treatment at Zadoks stage 13. **H.** Phenotypes under drought stress applied for 30 days at Zadoks stage 65. Scale bar: 10 cm. **I.** Soluble sugar accumulation in leaves at Fig 4H. **J.** GA content in spikelets and rachis of plants from Fig 4H. All data are presented as the mean ± SD unless otherwise stated (n ≥ 3 biological replicates).

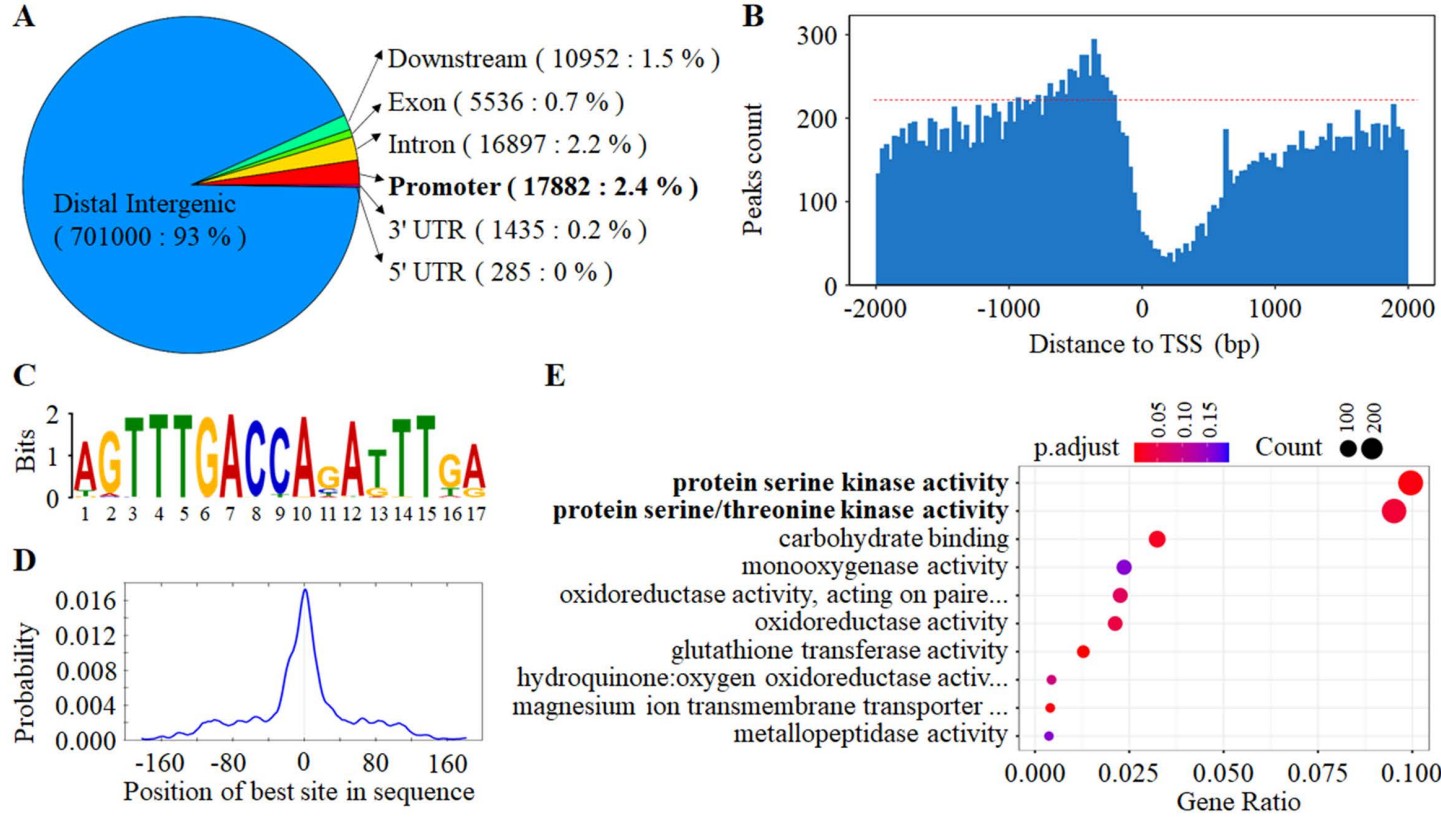

**Fig 5. Identification of TaWRKY58 target genes by DAP-seq. A.** Distribution of TaWRKY58 binding peaks across genomic features. **B.** Enrichment of TaWRKY58 peaks within ±1 kb of transcription start sites (TSS). **C.** De novo identified binding motif (Motif1) significantly enriched in TaWRKY58 peaks (E-value = 2.6$^{e-4164}$). **D.** Positional distribution frequency of Motif1 relative to peak centers. **E.** Top 10 enriched molecular functions among Motif1-containing target genes (GO analysis).

This suggests that the region mutated in mutant probe 2–1 contains the key target site for interaction with TaWRKY58. Thus, TaWRKY58 recognizes the sequence 'YVAAHYHKGKYYAAAYY,' with the W-box serving as the essential binding determinant.

We further employed the yeast one-hybrid system to verify the regulatory mechanism of *TaWRKY58* on *TaLRR* and *TaBCS1*. However, during the self-activation verification process, the TaWRKY58 sequence exhibited multiple self-activation regions (S4A Fig), and adding up to 80 mM 3-AT did not significantly inhibit the self-activation of the TaWRKY58 (S4B Fig). Therefore, we conducted transient expression assays. LUC assays demonstrated that TaWRKY58 significantly repressed the promoters of *TaLRR* and *TaBCS1* (Fig 6F and 6G). These results establish *TaWRKY58* as a transcriptional repressor that directly binds to W-box-containing promoters to suppress downstream gene expression.

### *TaWRKY58* negatively regulates *TaLRR* and *TaBCS1* to modulate plant height and drought response in wheat

To further elucidate the regulatory mechanism of *TaWRKY58* on *TaLRR* and *TaBCS1*, we obtained premature termination mutants for both genes. Through screening of a *Triticum aestivum* cv. Jing411 (J411) TILLING population, we identified one premature stop codon in *TaLRR* (p.Gln30*, Δ*talrr*) and one in *TaBCS1* (p.Trp200*, Δ*tabcs1*). Sanger sequencing confirmed that these mutations result in the truncation of over 50% of the corresponding protein sequences (Fig 7A and 7B). To minimize background mutations from chemical mutagenesis, homozygous Δ*talrr* and Δ*tabcs1* mutants were

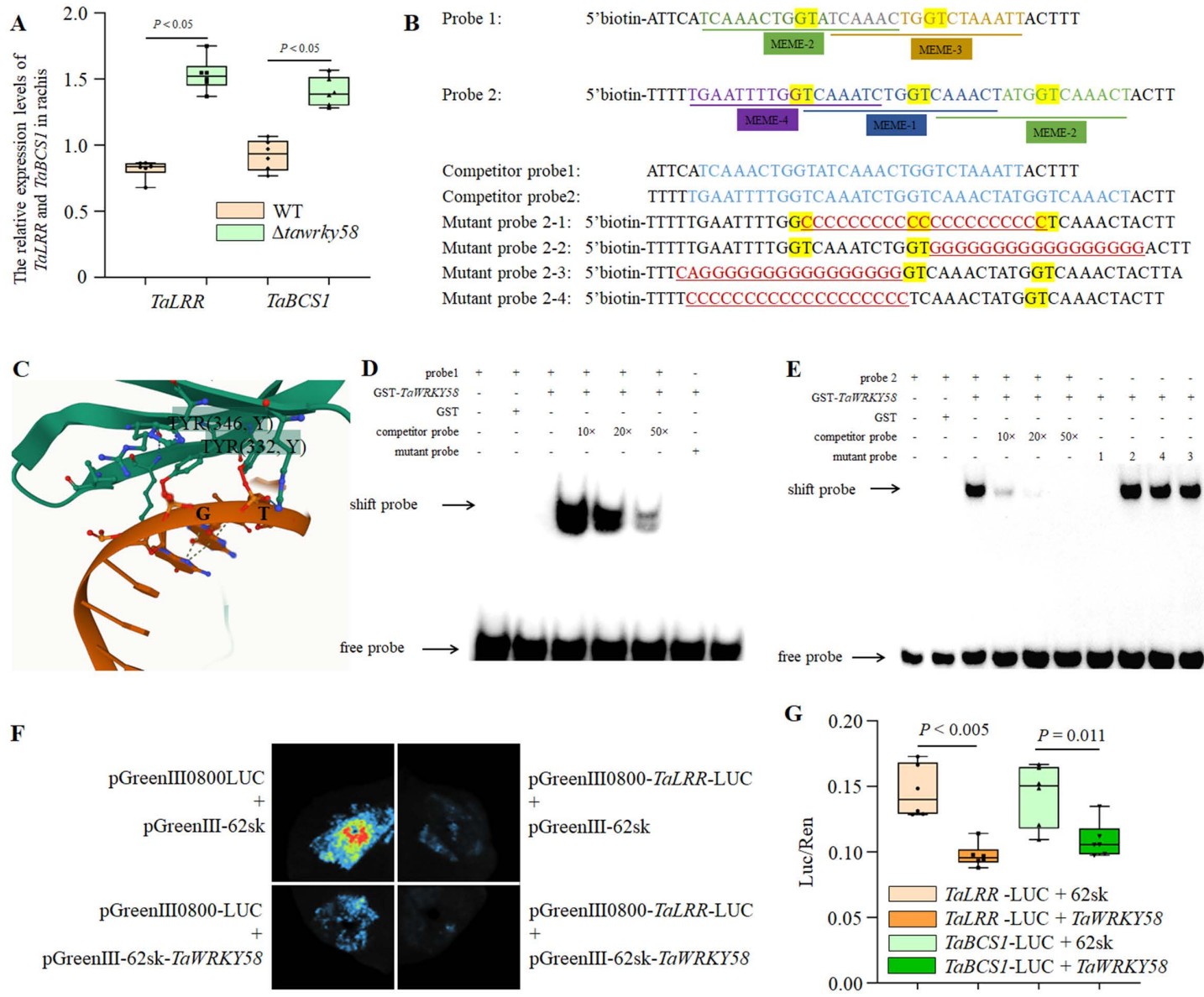

**Fig 6. Molecular mechanism of *TaWRKY58*-mediated transcriptional repression. A.** Relative expression levels of *TaLRR* and *TaBCS1* in the rachis of WT Δ*tawrky58* mutant at Zadoks stage 60, six biological replicates were performed for each plant. **B.** Schematic diagram of EMSA probes design. Probes were biotin labeled. Competition experiments were carried out by adding excessive unlabeled probes. **C.** The key interaction sites between TaW-RKY58 and motif1 were predicted through AlphaFold. **D.** EMSA showing recombinant TaWRKY58 binding to TaLRR promoter. Competitor: unlabeled probe. **E.** EMSA of TaWRKY58 binding to TaBCS1 promoter and mutant probes (M1-M4). Underline indicates binding loss with M1. **F.** Dual-LUC assay in N. benthamiana: 35S::TaWRKY58 represses proTaLRR::LUC. Vectors: pGreen0800-proTaLRR::LUC+pGreen62sk (± TaWRKY58). **G.** Normalized LUC/REN ratios in rice protoplasts confirming repression of proTaLRR::LUC and proTaBCS1::LUC by TaWRKY58, six biological replicates were performed for each plant.

backcrossed twice to the wild-type parental line Jing411. Phenotypic analysis revealed that although the Δ*talrr* and Δ*tabcs1* mutants did not exhibit alterations in flowering time, the Δ*tabcs1* mutant showed a significant reduction in plant height (Fig 7C and 7D). Under drought stress, the expression levels of *TaLRR* and *TaBCS1* were significantly upregulated

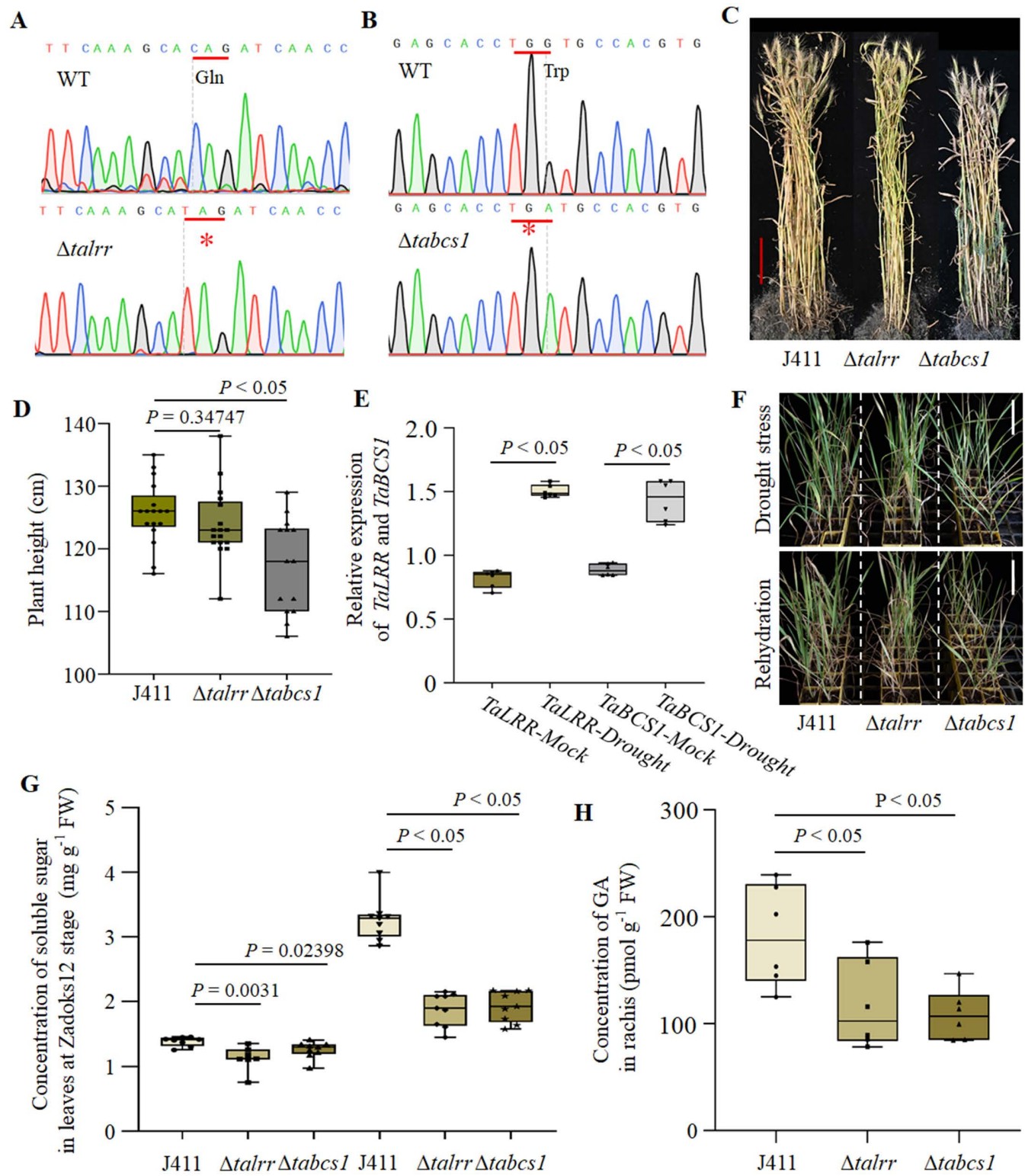

**Fig 7. Functional analysis of *TaLRR* and *TaBCS1* in wheat. A-B.** Sanger sequencing chromatograms confirming the mutation sites in the Δ*talrr* (A) and Δ*tabcs1* (B) mutants. **C.** Phenotypic comparison of wild-type (J411), Δ*talrr*, and Δ*tabcs1* plants at Zadoks growth stage 19. **D.** Plant height measurements of the indicated genotypes at Zadoks growth stage 65. **E.** the expression levels of *TaLRR* and *TaBCS1* in J411 leaves after 7 days of 20%

PEG6000 treatment at Zadoks stage 13. **F.** Phenotypes under drought stress (30 days) followed by rehydration for 12 days. **G.** Soluble sugar content in leaves of each genotype after 7 days of 20% (w/v) PEG6000 treatment at Zadoks growth stage 13. **H.** GA content in rachis of plants at Fig 7F. All data are presented as the mean ± SD unless otherwise stated (n ≥ 3 biological replicates).

under PEG6000-induced drought stress (Fig 6E). Both mutants displayed markedly enhanced sensitivity to drought, accompanied by a significant decrease in soluble sugar content under prolonged drought treatment (Fig 7F and 7G). And, GA content was significantly reduced in the Δ*talrr* and Δ*tabcs1* mutants under these conditions (Fig 7H). Collectively, these findings support a model wherein *TaWRKY58* coordinately regulates plant height and drought sensitivity by repressing the expression of *TaBCS1* and *TaLRR*, respectively.

## Discussion

Balancing growth with stress adaptation represents a fundamental constraint on wheat productivity. TFs, particularly those in the WRKY family, act as central regulators that integrate these competing processes [1,14]. While some WRKY members, such as *TaWRKY2* and *TaWRKY17*, enhance both stress tolerance and yield simultaneously [9,10], others exert more specialized roles. Here, we demonstrate that *TaWRKY58* coordinates this critical balance by functioning as a transcriptional repressor directly targeting *TaLRR* and *TaBCS1* (Fig 6). The increased plant height, accelerated flowering, and rachis-specific GA accumulation observed in the Δ*tawrky58* mutant (Fig 3D and 3I), together with the dwarf and drought-hypersensitive phenotypes of the Δ*talrr* and Δ*tabcs1* mutants (Fig 7D–7G), firmly establish a linear regulatory pathway of TF→target gene→phenotype. A key and paradoxical finding is that although *TaWRKY58* is strongly induced by multiple stresses (Fig 1B–1F), its loss of function did not markedly alter resistance to FHB, salt, or drought (Fig 4A–4H). This strongly suggests that *TaWRKY58* does not act as a direct mediator of canonical stress defense, but rather serves as a master regulator of resource allocation. Further supporting this role, expression analysis revealed that genes associated with flowering (e.g., *VRN1*) and GA biosynthesis (e.g., *GA20ox2*) were significantly upregulated in the Δ*tawrky58* mutant (S6 Fig) [15,16]. Similarly, under drought conditions, stress-responsive genes such as *TaABI5* and *TaLTP1* were also markedly induced in the mutant background (S7 Fig) [17,18]. Therefore, *TaWRKY58* likely mediates through the *TaWRKY58–TaLRR/TaBCS1* regulatory module to influence the expression of flowering and drought resistance–related genes, thereby integrating developmental signals with energy metabolism to finely tune the balance between growth and stress adaptation in wheat (Fig 8).

The enhanced drought tolerance observed in the Δ*tawrky58* mutant is substantiated by multiple lines of evidence. Soluble sugar accumulation serves as a reliable biochemical indicator of osmotic adjustment capacity, which is mechanistically linked to improved water retention and membrane stability under drought stress [19,20]. The increased cell viability

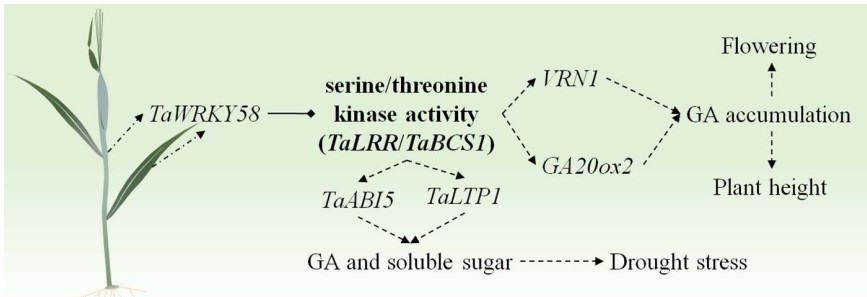

**Fig 8. Schematic model of *TaWRKY58*-mediated negative regulation of flowering and drought stress response in wheat.**

observed in Δ*tawrky58*provides cytological evidence for reduced tissue damage, functionally analogous to electrolyte leakage measurements (S5 Fig). Together, these data support a model where *TaWRKY58*-mediated repression restricts osmotic adjustment under non-stress conditions to prioritize growth.

The rapid yet transient induction of *TaWRKY58* by stress (Fig 1B-1D) suggests its role as an early "molecular sentinel". However, the lack of a strong stress susceptibility phenotype in the loss-of-function mutant (Fig 4) indicates that its primary function is not in mounting an acute defense, but rather in fine-tuning the balance between growth and stress-related metabolism, a phenomenon that can be described as induction without a direct role in immunity. We propose two non-mutually exclusive explanations for this observation. First, *TaWRKY58* induction may serve a priming function, preparing the plant to potentially curtail growth if a stress persists, rather than directly activating classical defense genes. Second, the extensive regulatory network governed by *TaWRKY58*, as evidenced by its thousands of potential targets identified by DAP-seq (Fig 5A and S3 Table), suggests that its function is likely buffered by genetic redundancy, where the loss of TaWRKY58 can be partially compensated by other TFs. Critically, the significant accumulation of soluble sugars in the Δ*tawrky58* mutant under drought (Fig 4G and 4I) provides direct metabolic evidence for its role in resource partitioning. This indicates that TaWRKY58-mediated repression is crucial for prioritizing growth over osmotic investment under non-stress conditions. In its absence, carbon resources are diverted into osmoprotectant synthesis even without a sustained stress signal. The expression patterns of its target genes under stress further support this model. While *TaWRKY58* is induced, its targets *TaLRR* and *TaBCS1* are ultimately upregulated under prolonged drought in wild-type plants (Fig 7E). This suggests that the initial, transient repression by *TaWRKY58* is eventually overridden by stronger, direct stress-responsive transcriptional activators, a classic feature of a regulatory module that integrates multiple signals to fine-tune outputs. Therefore, we hypothesize that the "priming" effect is molecularly manifested as a transient repression of targets like *TaBCS1* and *TaLRR*, which, when sustained, would limit growth. Although detailed kinetic analysis of this repression and de-repression cycle is an important avenue for future research, the steady-state changes in our mutant lines firmly establish *TaWRKY58* as a key repressor within this integrated network that coordinates growth and stress adaptation via the *TaWRKY58*-TaLRR/TaBCS1 module (Fig 8).

To further substantiate the link between these targets and hormone metabolism, we measured GA levels in the Δ*talrr* and Δ*tabcs1* mutants. Both mutants exhibited significantly reduced GA accumulation under drought stress (Fig 7H), consistent with their dwarf phenotypes and heightened drought sensitivity. This suggests that *TaWRKY58*-mediated repression of *TaLRR* and *TaBCS1* converges on modulating GA homeostasis, likely through distinct pathways: *TaLRR* (encoding a leucine-rich repeat protein) may positively participate in GA signaling or feedback regulation [21], whereas *TaBCS1* (encoding a mitochondrial AAA$^+$ATPase) could influence GA biosynthesis by affecting cellular energy status. The reduced soluble sugar accumulation in Δ*tabcs1* (Fig 7G) further supports a role for *TaBCS1* in energy metabolism that impacts both osmolyte production and hormone synthesis [22]. Future studies investigating GA pathway gene expression in Δ*talrr* and mitochondrial function in Δ*tabcs1* will help elucidate these mechanistic links.

In conclusion, we have delineated a coherent regulatory module in which *TaWRKY58* represses *TaLRR* and *TaBCS1* to calibrate plant height and metabolic responses to drought (Fig 8). The drought hypersensitivity and reduced solute accumulation in the loss-of-function mutants of these target genes confirm that this pathway integrally connects development with stress-associated metabolism (Fig 7F and 7G). The distinct molecular functions of the targets—cell signaling (*TaLRR*) and energy metabolism (*TaBCS1*)—provide a mechanistic framework for how a single transcription factor can achieve this multi-faceted regulation. Importantly, mutations in these three genes will not affect the thousand kernel weight, grain length and grain width of wheat (S8 Fig). Thus, our study suggests that fine-tuning this pathway, rather than complete disruption or overexpression, is key to improving crop resilience. Promising strategies include: 1) using CRISPR-based promoter engineering to modulate (not knockout) *TaWRKY58* expression levels, potentially by editing its auto-regulatory W-box elements to create alleles with weaker repressive activity; and 2) exploiting natural variation in the promoters of *TaLRR* and *TaBCS1*or coding sequences of *TaWRKY58* in breeding programs (S9 Fig). By partially releasing the brake

this module imposes on growth under stress, we may be able to breed wheat cultivars that maintain better yield stability under fluctuating environments without compromising intrinsic stress awareness.

## Materials and methods

### Plant materials and growth conditions

All mutants used in this study were generated in the background of durum wheat (*Triticum turgidum* ssp. *durum*) cv. 'Kronos' (AABB, 2n = 4x = 28). The early termination mutant for *TraesCS1A02G070400* (T4-2223, Δ*tawrky58*) was derived from an ethyl methanesulfonate (EMS)-mutagenized population of 'Kronos' [23]. Mutants for *TraesCS5A02G528800* (ID 5510427_5A_690570317_E_c.88C > T p.Gln30*, Δ*talrr*) and *TraesCS7B02G342600* (ID8278526_7B_602343351_E_c.6 00G > A p.Trp200*, Δ*tabcs1*) were obtained from an EMS-mutagenized population of Jing411 [24]. Transgenic plants were generated as described previously [25]. T$_1$ transgenic lines were verified using the primer pair Ubi1899F + Bar496R (S1 Table). Phenotypic analyses were conducted on the T$_2$ generation, and results were confirmed using the T$_3$ generation. All plants were grown in greenhouses under a 16/8 h day/night photoperiod at 25/16 °C. Plants were watered as needed and fertilized before sowing with 15-15-15 (N-P-K) fertilizer as basal fertilizer.

### Sequence analysis and primer design

The nucleotide sequence of *TaWRKY58* was retrieved from the Jorge Dubcovsky lab database (https://dubcovskylab.ucdavis.edu/). Its coding sequence was predicted using the *Triticum turgidum* Svevo.v1 genome database (http://plants.ensembl.org/index.html) and subsequently confirmed by Sanger sequencing. PCR primers were designed using Primer Premier 5.0 software (Premier Biosoft, Palo Alto, Canada. Deduced amino acid sequences were aligned using MEGA version 11 [26]. Neighbor-joining phylogenetic trees were constructed in MEGA using Poisson correction and complete gap deletion, with branch support assessed by 10,000 bootstrap replicates. Multiple sequence alignments were performed using DNAman 9.0 (Lynnon Biosoft, San Ramon, CA, USA). Protein domain prediction was conducted using UniProt (https://www.uniprot.org/), SMART (https://smart.embl.de/), and MobiDB (https://mobidb.org/). Intrinsically Disordered Regions (IDRs) were predicted using IUPred2 (https://iupred2a.elte.hu/).

### Gene expression analysis

Total RNA was extracted from freshly powdered wheat spikelets using the E.Z.N.A Total RNA Kit I (Omega Bio-Tek, Norcross, GA, USA) according to the manufacturer's instructions. RNA was reverse transcribed to cDNA using the Prime-Script RT Reagent Kit with gDNA Eraser (Takara, Dalian, China) following the manufacturer's protocol. Quantitative PCR (qPCR) was performed using the *w-GAPDH* (*Ta.66461*) and *Aox* (*Ta.6172*) genes as reference genes [11]. Reactions were carried out on a MyiQ Real-Time PCR Detection System (Bio-Rad, Hercules, CA, USA). All primers used are listed in S1 Table.

### Subcellular localization

Subcellular localization of TaWRKY58 was predicted using Cell-Ploc 2.0 (http://www.csbio.sjtu.edu.cn/bioinf/Cell-Ploc-2/). The *TaWRKY58* coding sequence (without stop codon) was fused in-frame to the N-terminus of GFP in the 163-*hGFP* vector (ProBUI::TaWRKY58-GFP) via LR recombination (Invitrogen). The fusion construct was transformed into 'Kronos' protoplasts following established protocols [27]. Subcellular localization of TaWRKY58-GFP was visualized using a Leica confocal microscope (Leica Microsystems Trading Co., Ltd., Shanghai, China).

### DNA-affinity purification sequencing (DAP-seq) and data analysis

DAP-seq was performed using genomic DNA from the hexaploid wheat cultivar 'Fielder' (AABBDD) to capture binding sites across all three subgenomes, including the D genome present in the hexaploid mutant 'Jing411'. Given that

WRKY DNA-binding domains recognize highly conserved W-box motifs across wheat cultivars, this approach allows genome-wide identification of *TaWRKY58* binding sites while maintaining alignment compatibility with the 'Chinese Spring' reference genome. Total genomic DNA was extracted from fresh leaves of wheat cv. 'Fielder' using the cetyl-trimethylammonium bromide (CTAB) method [28]. DNA was dissolved in 50 µL Tris-EDTA buffer (TE; 10 mM Tris-HCl, 1 mM EDTA, pH 8.0). Genomic DNA (5 µg in 130 µL TE) was fragmented to an average size of 200 bp using a Covaris M220 (Woburn, MA, USA) with manufacturer-recommended settings. Fragmented DNA was purified using MICH DNA Clean Beads (Cat# NGS0201; Bluescape Hebei Biotech Co., Ltd., Baoding, China) at a DNA:beads ratio of 0.7:1. Following a 5-min incubation at room temperature, beads were immobilized using a Magpow magnet (Bluescape). After supernatant removal, beads were washed twice with 200 µL 80% ethanol, air-dried, and resuspended in 22 µL resuspension buffer. After a 5-min incubation, beads were re-immobilized and the DNA-containing supernatant was collected. Libraries were constructed using the NEXTflex Rapid DNA-Seq Kit 2.0 (Revvity Health Sciences) per manufacturer's instructions. The *TaWRKY58* coding sequence was cloned into the pFN19K HaloTag T7 SP6 Flexi vector. HaloTag-TaWRKY58 fusion protein was expressed in the TNT SP6 High-Yield Wheat Germ Protein Expression System (Promega) in 50 µL reactions incubated at 37°C for 2 h. Expressed proteins were captured using Magne HaloTag Beads (Promega). DAP-seq binding assays, sequencing, and primary data analysis were performed by Bluescape Hebei Biotech Co. Ltd. (Baoding, China).

Raw sequencing data were processed using fastp with default parameters to obtain clean reads [29]. Clean reads were aligned to the Triticum aestivum reference genome (IWGSC RefSeq v2.1; EnsemblPlants) using BWA-MEM [30]. Mapped reads were filtered for MAPQ ≥ 30 using SAMtools to retain uniquely mapped reads [31]. Peak calling was performed using MACS2 and Homer, with peaks from technical replicates merged at Q < 0.05 [32,33]. De novo motif discovery was conducted with MEME-ChIP [34]. Peak annotation was performed using ChIPseeker [35].

## Transcriptional activation assay in yeast

The coding sequence of *TaWRKY58* and its truncated fragments were amplified from 'Kronos' cDNA, digested with *EcoRI* and *BamHI*, and cloned into the pGBKT7 vector. Recombinant plasmids were transformed into yeast strain $Y_2H$ Gold (WeiDi Biotech, Shanghai, China) using the PEG/LiAc method. Transformants were selected on synthetic defined (SD) medium lacking leucine and tryptophan (SD/-Leu/-Trp) and incubated at 30°C for 2–3 days. Positive recombinant clones were confirmed by PCR using Taq Plus PCR Mix (Biomed, Beijing, China). For transcriptional activation assays, yeast strains were cultured on SD medium lacking adenine, histidine, leucine, and tryptophan (SD/-Ade/-His/-Leu/-Trp) supplemented with 20–80 mM 3-aminotriazole (3-AT). Growth was assessed after 3 days at 30°C. Controls included pGBKT7-GAL4 (positive control) and empty pGBKT7 vector (negative control).

## Transient dual-luciferase (LUC) reporter assay

The promoters of *TaLRR* and *TaBCS1* were amplified from genomic DNA, digested with *HindIII* and *SmaI*, and cloned into the pGreenIII 0800-LUC vector to generate proTaLRR::LUC and proTaBCS1::LUC reporter constructs. The *TaWRKY58* coding sequence was digested with *SacI* and *SpeI* and cloned into the pGreenIII 62-SK vector to create the effector construct (35S::*TaWRKY58*).

Protoplast cotransfection was performed as described previously [27]. Luciferase activities were measured 16–24 h post-transfection using the TransDetect Double-Luciferase Reporter Assay Kit (TransGen Biotech, Beijing, China) according to the manufacturer's protocol. Luminescence was quantified using a Varioskan LUX microplate reader (Thermo Scientific, MA, USA). Relative luciferase activity was calculated as the ratio of firefly luciferase (FLUC) to Renilla luciferase (RLUC) activity (FLUC/RLUC).

## Electrophoretic mobility shift assay (EMSA)

EMSA was performed using the EMSA Probe Biotin Labeling Kit and Chemiluminescent EMSA Kit (Beyotime, Shanghai, China) according to the manufacturer's protocol. Recombinant MBP-*TaWRKY58* and MBP (negative control) proteins were purified from *E. coli* BL21 (DE3). Biotin-labeled DNA probes corresponding to the *TaLRR* and *TaBCS1* promoter regions were synthesized (sequences in S1 Table), with biotin conjugated to the 5' terminus (5'-ATGCATGCATGCAT GCATGCATGC). Unlabeled identical fragments served as competitors. Binding reactions contained 200 ng purified protein and 20 fmol biotinylated probe in 20 μL reaction buffer. After 30-min incubation at room temperature, samples were separated on 6% non-denaturing polyacrylamide gels in 0.5 × TBE buffer (44.5 mM Tris, 44.5 mM boric acid, 1 mM EDTA) at 60 V for 1 h at 4°C. Proteins were transferred to Biodyne B nylon membranes (Pall Corporation) and crosslinked. Signals were detected using the chemiluminescent substrate included in the kit and visualized with an Amersham Imager 600 (GE Healthcare, Amersham, UK). Images were acquired using a ChemiDoc XRS+ system (Bio-Rad Laboratories, Hercules, CA, USA).

## FHB resistance assay

FHB resistance assays were conducted at Zadoks growth stage 65 (Z65) using *F. graminearum* strain PH-1, a highly virulent wheat pathogen. Two florets of a central spikelet per head were point-inoculated with $1 \times 10^3$ conidia in 2 μL suspension. Inoculated heads were misted with sterile water, enclosed in plastic bags, and maintained at 25°C with ≥85% relative humidity (RH) for 48 h. Plants were subsequently transferred to a controlled-environment chamber at 25°C/60% RH under a 16/8 h day/night photoperiod. Disease symptoms were assessed 10 days post-inoculation (dpi) by counting infected spikelets. Twenty plants per line were evaluated, with all lines grown under identical greenhouse conditions (25/16°C day/night, 16/8 h photoperiod, ≥ 60% RH). Inoculated plants were maintained at ≥80% relative humidity. After 48 h, spikes exhibiting differential SA accumulation phenotypes were harvested, flash-frozen in liquid nitrogen, and ground to a fine powder. SA levels were quantified using monoclonal antibody-based ELISA kit (Ruixin Biological Technology, Quanzhou, China) according to the manufacturer's protocol. Analyses included three biological replicates per treatment.

## Salt and drought stress treatments

Healthy seeds were surface-sterilized with 75% ethanol, rinsed with sterile water, and germinated on moist filter paper in Petri dishes. Uniform seedlings at Zadoks growth stage 12 (Z12) were selected for salt (200 mM) and drought (20% (w/v) PEG6000) stress treatments. After 7 days of treatment, leaves were harvested for RNA extraction, ABA quantification, and soluble sugar measurement. Soluble sugar content was determined using the Plant Soluble Sugar Colorimetric Assay Kit (Elabscience Biotechnology, Wuhan, China). ABA and $GA_1$ content were measured by monoclonal antibody-based ELISA (Ruixin Biological Technology, Quanzhou, China), both following manufacturers' protocols.

Germinated seedlings were transplanted into 96-well plug trays (Spring Valley Agriscience, Jinan, China) and grown to Zadoks stage 19 (Z19) for potted plant treatments. For salt stress treatment, plants were irrigated with 2 L of 200 mM NaCl solution. Subsequent watering also used 200 mM NaCl solution. For drought stress treatment, no water was applied after the plants reached Zadoks stage 19. Unless specifically noted, all chemicals were purchased from VerSci (Stverbio, Beijing, China).

## Statistical analysis

Statistical analyses were performed using GraphPad Prism 8 (GraphPad Software, Boston, MA, USA). Statistical differences were analyzed using the least significant difference test at $p \leq 0.05$.

## Supporting information

**S1 Data. Underlying numerical data.**
(XLSX)

**S1 Fig. Phylogenetic analysis of TaWRKY58 protein sequences in wheat by Neighbor-joining tree.** The reported protein sequences and classification basis are as described previously [3]. The unreported protein sequences were obtained through Blast search of the reported protein sequences in EnsemblPlants (http://plants.ensembl.org/index.html), and all were confirmed through domain prediction. Bootstrap test of phylogeny was conducted with 10,000 replicates, and branch length values were displayed at the nodes of each proposed class.
(TIF)

**S2 Fig. Transcriptional expression patterns of TaWRKY58 gene during wheat development or upon treatments with various treatment.** A. The transcription levels (log2FC) of TaWRKY58 during wheat development based on public database (The Bio-Analytic Resource for Plant Biology: https://bar.utoronto.ca/). B. The transcription levels (log2FC) of TaWRKY58 under *Fusarium pseudograminearum* (Fp) and *Fusarium graminearum* (Fg) infection, NaCl, PEG6000, Phosphate (Pi), jasmonic acid (JA), abscisic acid (ABA) and salicylic acid (SA) stress based on WheatOmics 1.0 database [36].
(TIF)

**S3 Fig. Restriction analysis identified PCR products amplified by dCAPS markers in two homozygous genotypes (WT and Δ*tawrky58* mutant).** The red arrow represents Δ*tawrky58* mutant. The primers sequence listed in S1 Table.
(TIF)

**S4 Fig. Auto-activation assay of TaWRKY58.** A. Prediction of the self-activation region of TaWRKY58 and schematic diagram of the positions of different fragments. B. Auto-activation assay of TaWRKY58 in yeast.
(TIF)

**S5 Fig. The cross-section view of the tender part of the spike axis in Fig 4H.** Scale bar = 100 μm.
(TIF)

**S6 Fig. qPCR analysis of the expression levels of flowering- and GA biosynthesis-related genes in WT and Δ*tawrky58* mutants.** Asterisks (*) above bars indicate significant differences ($P \leq 0.05$) relative to controls (Student's t-test). Three biological replicates were performed for each plant.
(TIF)

**S7 Fig. qPCR analysis of the expression levels of drought stress-related genes in WT and Δ*tawrky58* plants.** Asterisks (*) above bars indicate significant differences ($P \leq 0.05$) relative to controls (Student's t-test). Three biological replicates were performed for each plant.
(TIF)

**S8 Fig. Effect of mutations in *TaWRKY58, TaLRR,* and *TaBCS1* on wheat yield-related traits.** Thousand kernel weight (A), grain length (B), and grain width (C) were measured for the indicated genotypes. No significant differences were observed between any of the mutants and their respective wild-type controls (Student's t-test, $P > 0.05$). Data are presented as mean ± SD (n ≥ 10).
(TIF)

**S9 Fig. Schematic of *TaWRKY58* (A), *TaLRR* (B), and *TaBCS1* (C) variations in wheat union database (http://wheat.cau.edu.cn/WheatUnion/).**
(TIF)

**S1 Table. List of all primers used in the experimental procedures.**
(XLSX)

**S2 Table. Genome-wide list of DNA sequences bound by TaWRKY58 as determined by DAP-seq analysis.**
(XLSB)

**S3 Table. Candidate target genes identified by the presence of motif1 binding sites in their promoter regions.**
(XLSX)

**S4 Table. GO terms significantly enriched among genes harboring motif1 in their promoters.**
(XLSX)

## Acknowledgments

We thank pro. Jorge Dubcovsky (UC Davis) and Francine Paraiso (UC Davis) for kindly provided 'Kronos' EMS mutants. We also thank pro. Luxiang Liu (Chinese Academy of Agricultural Science) for kindly provided 'Jing411' EMS mutants.

## Author contributions

**Conceptualization:** Yazhou Zhang, Yuming Wei.

**Formal analysis:** Xinyao Cheng, Xinyu Yu.

**Investigation:** Xinyao Cheng, Xinyu Yu, Anyu Gu, Xufei Zhao.

**Methodology:** Yazhou Zhang, Xinyao Cheng.

**Writing – original draft:** Yazhou Zhang.

**Writing – review & editing:** Yazhou Zhang, Mei Deng, Guoyue Cheng, Qiang Xu, Qiantao Jiang, Yuming Wei.

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
