## [Decision Letter · Decision Letter 0]

16 Feb 2026

PGENETICS-D-25-01211

The transcription factor TaWRKY58 coordinates growth and drought sensitivity in wheat by repressing TaLRR and TaBCS1

PLOS Genetics

Dear Dr. Zhang,

Thank you for submitting your manuscript to PLOS Genetics. After careful consideration, we feel that it has merit but does not fully meet PLOS Genetics's publication criteria as it currently stands. Therefore, we invite you to submit a revised version of the manuscript that addresses the points raised during the review process.

We look forward to receiving your revised manuscript.

Kind regards,

Angela Hancock

Section Editor

PLOS Genetics

Aimée Dudley

Editor-in-Chief

PLOS Genetics

Anne Goriely

Editor-in-Chief

PLOS Genetics

**Additional Editor Comments:**

Sorry that it took so long to get reviews back to you on this manuscript. I have not been able to find an Associate Editor willing to handle the manuscript and also had a very difficult time finding reviewers. Because of this, I am moving forward with the single review (Rev 2), but may try again to find additional reviewers if questions arise. Please address all comments by this reviewers. Thank you.

**Journal Requirements:**

At this stage, the following Authors/Authors require contributions: Yazhou Zhang, Xinyao Cheng, Xinyu Yu, Anyu Gu, Xufei Zhao, Mei Deng, Guoyue Cheng, Qiang Xu, Qiantao Jiang, and Yuming Wei. Please ensure that the full contributions of each author are acknowledged in the "Add/Edit/Remove Authors" section of our submission form.

The list of CRediT author contributions may be found here: https://journals.plos.org/plosgenetics/s/authorship#loc-author-contributions

- ® on pages: 18, 20, and 21

- TM on pages: 18, 19, 20, and 21.

Potential Copyright Issues:

i) Please confirm (a) that you are the photographer of 3C, 3F, 4A, 4E, 4H, 7C, and 7F, or (b) provide written permission from the photographer to publish the photo(s) under our CC BY 4.0 license.

ii) Figures S2, 2A, and 8. Please confirm whether you drew the images / clip-art within the figure panels by hand. If you did not draw the images, please provide (a) a link to the source of the images or icons and their license / terms of use; or (b) written permission from the copyright holder to publish the images or icons under our CC BY 4.0 license. Alternatively, you may replace the images with open source alternatives. See these open source resources you may use to replace images / clip-art:

1) Please clarify all sources of financial support for your study. List the grants, grant numbers, and organizations that funded your study, including funding received from your institution. Please note that suppliers of material support, including research materials, should be recognized in the Acknowledgements section rather than in the Financial Disclosure

2) State the initials, alongside each funding source, of each author to receive each grant. For example: "This work was supported by the National Institutes of Health (####### to AM; ###### to CJ) and the National Science Foundation (###### to AM)."

3) State what role the funders took in the study. If the funders had no role in your study, please state: "The funders had no role in study design, data collection and analysis, decision to publish, or preparation of the manuscript."

4) If any authors received a salary from any of your funders, please state which authors and which funders..

6) Your current Financial Disclosure states, "The author(s) received no specific funding for this work.".

However, your funding information on the submission form indicates receiving funds from National Key Research and Development Program-Creation and Application of New Disease-Resistant Wheat Germplasm in the Southwest Wheat-growing Area and Science and Technology Department of Sichuan Province.

Please indicate by return email the full and correct funding information for your study and confirm the order in which funding contributions should appear. Please be sure to indicate whether the funders played any role in the study design, data collection and analysis, decision to publish, or preparation of the manuscript.

**Reviewers' comments:**

Reviewer's Responses to Questions

**Comments to the Authors:**

Reviewer #2: This manuscript presents a well-structured and significant study. The authors convincingly identify TaWRKY58 as a key transcriptional repressor that fine-tunes the trade-off between plant height and drought response in wheat. The combination of genetic, molecular, and phenotypic analyses is a major strength. The proposed regulatory module is novel and has clear implications for wheat breeding. However, few comments listed in below:

1. While the direct repression of TaLRR and TaBCS1 by TaWRKY58 is solidly demonstrated, the functional link between these targets and the observed phenotypes could be more deeply explored. The manuscript states TaLRR is a signaling protein and TaBCS1 is an energy metabolism protein, but how exactly their repression by TaWRKY58 leads to altered GA levels (for TaLRR?) and soluble sugar accumulation (for TaBCS1?) remains somewhat speculative. Thus, it will be good if co-authors could add data or a more detailed discussion on: for example, Does TaLRR mutation affect GA biosynthesis/signaling genes or pathways? Does TaBCS1 mutation directly impact mitochondrial function (e.g., ATP levels, ROS) or the expression of sugar metabolism genes under drought? Measuring GA levels in the Δtahr mutant and soluble sugars/energy status in the Δtabcs1 mutant would significantly strengthen the causal chain from TF to target to metabolic phenotype.

2. The observation that TaWRKY58 is stress-induced but its loss-of-function does not confer strong stress susceptibility (Fig. 4) is intriguing and well-discussed. However, the proposed "priming" or "buffering" hypotheses, while reasonable, are not directly tested. It will be good if the co-authors could analyze the kinetics of target gene repression (TaLRR, TaBCS1) following stress induction of TaWRKY58. Does transient TaWRKY58 induction lead to sustained repression of its targets? This would provide molecular evidence for a priming mechanism.

3. The drought tolerance data for the Δtawrky58 mutant is based on soluble sugar accumulation and survival cell counts (S5 Fig). More comprehensive physiological data would bolster the claim of "enhanced drought tolerance." It will be good if co-authors could Include standard drought physiology metrics for the key genotypes (WT, Δtawrky58, overexpression line), such as: Water loss rate (relative water content over time), Stomatal conductance or photosynthetic parameters under progressive drought, or a clearer yield component assessment (e.g., spike fertility, grain number/weight) after drought-recovery cycles, especially given the focus on "yield stability."

4. The figure legends often state "n ≥ 3 biological replicates" or similar. For critical phenotypic measurements (e.g., plant height, soluble sugar content), the exact 'n' for each genotype should be explicitly stated in the legend or figure. The "Material and Methods" states two independent experimental replicates were conducted; this should be clarified in the context of biological vs. technical replication.

5. Fig 3H & 4J: Please specify which specific GA(s) were measured (e.g., GA1, GA4).

Fig 4: The legend for panels H, I, J references "figure 4H." This is confusing. Please correct the labels (likely should be "in plants from H" or similar).

Fig 6A: It would be helpful to show TaLRR and TaBCS1 expression in the overexpression line as well, to confirm the repressive effect.

Page 13, Line 200-201: "We detected 754,059 TaWRKY58-bound peaks" – This number seems extremely high. Please clarify if these are raw peaks from sequencing or after stringent filtering. A brief comment on what this high number might imply (e.g., low binding specificity, or a truly global regulator) would be useful.

Page 16, Line 283-284: "Under drought stress, the expression levels of TaLRR and TaBCS1 were significantly upregulated..." This seems counterintuitive if TaWRKY58 is induced and represses them. Please clarify or discuss this point—it might indicate a feedback loop or stress-induced override of repression.

6. The discussion is strong and thoughtful. The model (Fig. 8) is helpful but could be slightly enhanced. Consider adding the metabolic outputs (GA, soluble sugars) and the resulting phenotypes (height, drought tolerance) to the schematic to make the pathway from molecular mechanism to agronomic trait even clearer.

7. DAP-seq: The method states genomic DNA was from 'Fielder,' but the study uses 'Kronos' and 'Jing411' mutants. Please justify the use of a different cultivar for this genome-wide assay and discuss potential limitations.

**Have all data underlying the figures and results presented in the manuscript been provided?**

Large-scale datasets should be made available via a public repository as described in the *PLOS Genetics*
data availability policy, and numerical data that underlies graphs or summary statistics should be provided in spreadsheet form as supporting information., and numerical data that underlies graphs or summary statistics should be provided in spreadsheet form as supporting information., and numerical data that underlies graphs or summary statistics should be provided in spreadsheet form as supporting information., and numerical data that underlies graphs or summary statistics should be provided in spreadsheet form as supporting information.

Reviewer #1: Yes

Reviewer #2: Yes

PLOS authors have the option to publish the peer review history of their article (what does this mean?). If published, this will include your full peer review and any attached files.). If published, this will include your full peer review and any attached files.). If published, this will include your full peer review and any attached files.). If published, this will include your full peer review and any attached files.

...

Reviewer #1: No

Reviewer #2: No

**Figure resubmission:**
---

## [Decision Letter · Decision Letter 1]

7 Apr 2026

Dear Dr Zhang,

We are pleased to inform you that your manuscript entitled "The transcription factor TaWRKY58 coordinates growth and drought sensitivity in wheat by repressing TaLRR and TaBCS1" has been editorially accepted for publication in PLOS Genetics. Congratulations!

Yours sincerely,

Angela Hancock

Section Editor

PLOS Genetics

Aimée Dudley

Editor-in-Chief

PLOS Genetics

Anne Goriely

Editor-in-Chief

PLOS Genetics

BlueSky: @plos.bsky.social

Comments from the reviewers (if applicable):

Reviewer's Responses to Questions

**Comments to the Authors:**

Reviewer #2: The authors have provided a thorough, well-organized, and respectful response to the reviewers’ comments. They have addressed nearly all points with a combination of: new experimental data (e.g., GA measurements in Δt alrr and Δt abcs1 mutants), clarifications in figures and legends, revisions to the discussion to strengthen mechanistic interpretation and acknowledgment of limitations and future research directions. The response is scientifically sound and appropriate for a revision. Two minor comments issues need to clarify:

1. Fig. 8 revision: Updated to include metabolic outputs (GA, soluble sugars) and phenotypes – good improvement. However, Duplicate/redundant text (e.g., pages 73–74, 80–81) appears to be a formatting artifact from merging revisions. These do not affect scientific integrity but should be cleaned up in final proof.

2. Comment 5. The authors chose not to add O-TaWRKY58 expression data to Fig. 6A to avoid redundancy. This is acceptable for me, because the GA specification and legend corrections are appropriate. The decision not to add overexpression data to Fig. 6A is justified and does not weaken the conclusion.

**Have all data underlying the figures and results presented in the manuscript been provided?**

Large-scale datasets should be made available via a public repository as described in the *PLOS Genetics*
data availability policy, and numerical data that underlies graphs or summary statistics should be provided in spreadsheet form as supporting information., and numerical data that underlies graphs or summary statistics should be provided in spreadsheet form as supporting information., and numerical data that underlies graphs or summary statistics should be provided in spreadsheet form as supporting information., and numerical data that underlies graphs or summary statistics should be provided in spreadsheet form as supporting information.

Reviewer #2: Yes

PLOS authors have the option to publish the peer review history of their article (what does this mean?). If published, this will include your full peer review and any attached files.). If published, this will include your full peer review and any attached files.). If published, this will include your full peer review and any attached files.). If published, this will include your full peer review and any attached files.

...

Reviewer #2: No

**Data Deposition**

If you have submitted a Research Article or Front Matter that has associated data that are not suitable for deposition in a subject-specific public repository (such as GenBank or ArrayExpress), one way to make that data available is to deposit it in the Dryad Digital Repository. As you may recall, we ask all authors to agree to make data available; this is one way to achieve that. A full list of recommended repositories can be found on our . As you may recall, we ask all authors to agree to make data available; this is one way to achieve that. A full list of recommended repositories can be found on our . As you may recall, we ask all authors to agree to make data available; this is one way to achieve that. A full list of recommended repositories can be found on our . As you may recall, we ask all authors to agree to make data available; this is one way to achieve that. A full list of recommended repositories can be found on our website....

http://datadryad.org/submit?journalID=pgenetics&manu=PGENETICS-D-25-01211R1

Additionally, please be aware that our data availability policy requires that all numerical data underlying display items are included with the submission, and you will need to provide this before we can formally accept your manuscript, if not already present. requires that all numerical data underlying display items are included with the submission, and you will need to provide this before we can formally accept your manuscript, if not already present. requires that all numerical data underlying display items are included with the submission, and you will need to provide this before we can formally accept your manuscript, if not already present. requires that all numerical data underlying display items are included with the submission, and you will need to provide this before we can formally accept your manuscript, if not already present.

**Press Queries**

If you or your institution will be preparing press materials for this manuscript, or if you need to know your paper's publication date for media purposes, please inform the journal staff as soon as possible so that your submission can be scheduled accordingly. Your manuscript will remain under a strict press embargo until the publication date and time. This means an early version of your manuscript will not be published ahead of your final version. PLOS Genetics may also choose to issue a press release for your article. If there's anything the journal should know or you'd like more information, please get in touch via plosgenetics@plos.org....

---

## [Editor Report · Acceptance letter]

PGENETICS-D-25-01211R1

The transcription factor TaWRKY58 coordinates growth and drought sensitivity in wheat by repressing TaLRR and TaBCS1

Dear Dr Zhang,

We are pleased to inform you that your manuscript entitled "The transcription factor TaWRKY58 coordinates growth and drought sensitivity in wheat by repressing TaLRR and TaBCS1" has been formally accepted for publication in PLOS Genetics! Your manuscript is now with our production department and you will be notified of the publication date in due course.

With kind regards,

Anita Estes

PLOS Genetics

On behalf of:
